

# Breviscapine alleviates podocyte injury by inhibiting NF-κB/NLRP3-mediated pyroptosis in diabetic nephropathy

Linlin Sun, Miao Ding, Fuhua Chen, Dingyu Zhu and Xinmiao Xie

Department of Nephrology, Tongren Hospital Shanghai Jiaotong University School of Medicine, Shanghai, China

## ABSTRACT

Podocyte injury is a critical factor in the pathogenesis of diabeticnephropathy (DN). Emerging evidence has demonstrated that breviscapine (Bre) exerts a renoprotective effect on diabetic rats. However, the effects of Bre on regulating podocyte injury under high glucose (HG) conditions remain unclear. In this study, an experimental mouse model of DN was induced by intraperitoneal injections of streptozotocin (STZ) *in vivo*. The effects of Bre on podocyte injury were assessed using cell counting kit-8 (CCK-8) assay, TdT-mediated dUTPnick-endlabelling (TUNEL) staining, quantitative real-time PCR (qRT-PCR) and western blot analysis. We found that renal function was significantly decreased in diabetic mice, and this effect was blocked by Bre treatment. Bre effectively increased podocyte viability and inhibited HG-induced cell apoptosis. Furthermore, Bre ameliorated HG-induced podocyte injury, as evidenced by decreased α-smooth muscle actin (α-SMA) expression and increased podocin and synaptopodin expression. Mechanistically, Bre inhibited HG-induced nuclear factorkappaB (NF-κB) signalling activation and subsequently decreased NLR family pyrin domain containing 3 (NLRP3) inflammasome activation, resulting in a decrease in pyroptosis. Pharmacological inhibition of NLRP3 decreased HG-induced podocyte injury, whereas the NLRP3 agonist abrogated the effects of Bre on inhibiting podocyte injury. In summary, these results demonstrate that Bre alleviates HG-induced podocyte injury and improves renal function in diabetic mice, at least in part by inhibiting NF-κB/NLRP3-mediated pyroptosis.

## INTRODUCTION

Diabetic nephropathy (DN) is a serious diabetic microvascular complication that occurs in up to 20–50% of individuals with diabetes (*Alicic, Rooney & Tuttle, 2017*; *Selby & Taal, 2020*). The morphological alterations in glomeruli and tubules caused by podocyte injury and excessive mesangial cell proliferation are the two major characteristics of DN (*Nagata, 2016*; *Rousseau et al., 2022*; *Thomas & Ford Versypt, 2022*). In addition to controlling blood glucose and blood pressure, blocking the renin-angiotensin-aldosterone system is a therapeutic option for DN (*Sawaf et al., 2022*). However, these treatments cannot reverse

Corresponding author
Linlin Sun, llsunzj@sina.com

this disease. It is critical to uncover the underlying mechanisms of podocyte injury to develop an effective therapy.

Podocytes, which are glomerular visceral epithelial cells, are essential components of the renal filtration barrier that create slit diaphragms within the glomeruli (*Shankland, 2006*; *Yan et al., 2022*). Mounting evidence has shown that podocyte injury is an early and critical event in the pathogenesis of DN (*Asanuma, 2015*; *Nagata, 2016*; *Yasuno et al., 2010*). Hyperglycaemia leads to podocyte injury through various pathways, including inflammation, reactive oxygen species (ROS), and endoplasmic reticulum stress (*Schena & Gesualdo, 2005*; *Wolf, 2004*). These factors result in podocyte apoptosis, aberrant expression of interconnecting slit diaphragm proteins (nephrin and podocin), and cytoskeletal rearrangement in podocytes (*El-Aouni et al., 2006*; *Jefferson, Shankland & Pichler, 2008*).

Intracellular inflammation can be activated by various stimuli and is a typical mechanism by which podocyte damage occurs (*Wang et al., 2022a*). The NLRP3 inflammasome has critical effects on the production of proinflammatory cytokines, and is closely correlated with podocyte damage and DN (*Jiang et al., 2021*; *Xu et al., 2021*; *Zhang et al., 2022a*). NLRP3 inflammasome activation in podocytes accelerates kidney injury in DN (*Shahzad et al., 2022*), whereas CY-09, a specific NLRP3 inflammasome antagonist, alleviates renal damage in DN by repressing NLRP3 activation (*Yang & Zhao, 2022*). Moreover, NLRP3 inflammasome-induced pyroptosis is a new form of cell death associated with podocyte injury (*Lin et al., 2020*). As an intracellular pattern recognition receptor (PRR), NLRP3 can be activated by many pathogen-associated molecular patterns (PAMPs) and damage-associated molecular patterns (DAMPs). Subsequently, NLRP3 assembles with the adaptor ASC (apoptosis-associated speck-like protein) and pro-caspase-1, resulting in ASC oligomerization and caspase-1 activation. Active caspase-1 catalyses IL-18 and IL-1β maturation (*Lasithiotaki et al., 2018*; *Peng et al., 2020*) and cleaves gasdermin D (GSDMD) into the N-terminal (GSDMD-N) and C-terminal (GSDMD-C) forms (*Ding et al., 2016*; *Latz, Xiao & Stutz, 2013*). GSDMD-N forms pores in the cell membrane and mediates perforation, which results in pyroptosis and proinflammatory factor release (*Shi et al., 2015*). Pharmacological inhibition of NLRP3-mediated pyroptosis can effectively alleviate podocyte injury in DN (*Feng et al., 2021*; *Wang et al., 2022a*; *Zhang et al., 2022a*).

Breviscapine (Bre), a flavonoid extract from *Erigeron breviscapus*, exerts renoprotective effects on diabetic rats. Bre treatment decreases albuminuria and alleviates glomerular hypertrophy and tubulointerstitial injury (*Qi et al., 2006*; *Xu et al., 2013*). Bre alleviates renal fibrosis and contrast medium-induced nephropathy (*Jiang et al., 2016*). Based on above findings, we speculated whether Bre relieves HG-induced podocyte injury and improves renal function in diabetic mice by regulating NLRP3 inflammasome.

## MATERIALS AND METHODS

### Animal model

This work was performed with the approval of the Animal Ethics Committee of Shanghai Jiao Tong University Affiliated Tong Ren Hospital (No. 2019-060) according to the

ARRIVE guidelines to minimize animal suffering (*du Sert et al., 2020*). Male C57BL/6J wild-type mice (approximately 8 weeks old) were purchased from Shanghai Regan Biotechnology (Shanghai, China), housed in a thermostatic room (25 ± 2 °C) with 12/12 light/dark cycles, and given free access to chow and water. The mice were randomly divided into three groups (*n* = 7): control group, diabetic group, and Bre-treated group. Diabetes was induced by intraperitoneal (i.p.) injections of STZ (Aladdin, Shanghai, China) at a dose of 50 mg/kg for 5 consecutive days, as previously described (*Lee et al., 2022*; *Song et al., 2022*; *Zhong et al., 2019*). Mice with blood glucose >16.7 mmol/L were considered diabetic 3 days after the injection (*He et al., 2013*). Based on the previous studies (*Jiang et al., 2016*; *Lan et al., 2022*) and our preliminary experiments, diabetic mice in the Bre-treated group were administered Bre daily by oral gavage at a dose of 15 or 30 mg/kg for 4 weeks. Bre was purchased from Shanghai Winherb Medical Science Co., Ltd., (Shanghai, China). Control mice were i.p. administered sodium citrate buffer. After being treated, all mice were euthanized by cervical dislocation under deep anaesthesia (pentobarbital sodium, 40 mg/kg), and renal tissues and blood samples were collected to carry out biochemical and pathological assessments, respectively. The establishment of diabetic mice model and treatment procedure were displayed in Fig. 1A. In this study, animals suffering from unexpected disease were euthanized as described previously.

## Cell culture and treatments

The immortalized mouse podocyte line MPC5 was purchased from the Cell Bank of the Chinese Academy of Sciences (Shanghai, China). To induce differentiation, MPC5 podocytes were cultured in RPMI-1640 medium (Gibco, New York, NY, USA) supplemented with 10% FBS (Gibco, New York, NY, USA) without IFN-γ for 2 weeks in a 5% $CO_2$ incubator (*Liu et al., 2016a*). Differentiated MPC5 cells were used in subsequent experiments. MPC5 cells were treated with HG (30 mM) to establish an *in vitro* DN model, and 5 mM glucose was used as a control (low glucose). To examine the effects of Bre, MCC950, nigericin, and Ac-YVAD-cmk on MPC5 cell injury, the cells were treated with 50 μM or 100 μM Bre, 10 μM MCC950 (Merck, Rahway, MA, USA), 10 μM nigericin (MCE, Princeton, NJ, USA), and 40 μM Ac-YVAD-cmk (Merck, Rahway, MA, USA), and then cell injury was assessed.

## qRT–PCR

Total RNA was isolated from MPC5 cells or renal tissues with TRIzol reagent (TaKaRa, Tokyo, Japan), quantified using a Nanodrop (Thermo Fisher Scientific, Waltham, MA, USA), and then used to synthesize first-strand cDNA using M-MLV (TaKaRa, Tokyo, Japan) in a final volume of 20 μl. qRT–PCR was performed in triplicate using KAPA SYBR® FAST (Roche, Basel, Switzerland) on a Bio-Rad CFX96 qPCR System (CA, USA) under the following conditions: 94 °C for 10 min, followed by 38 cycles (94 °C for 15 s and 59 °C for 15 s). Gene expression was calculated using the $2^{(-\Delta\Delta CT)}$ method after normalization to β-actin (*Livak & Schmittgen, 2001*). All primers are shown in Table S1.

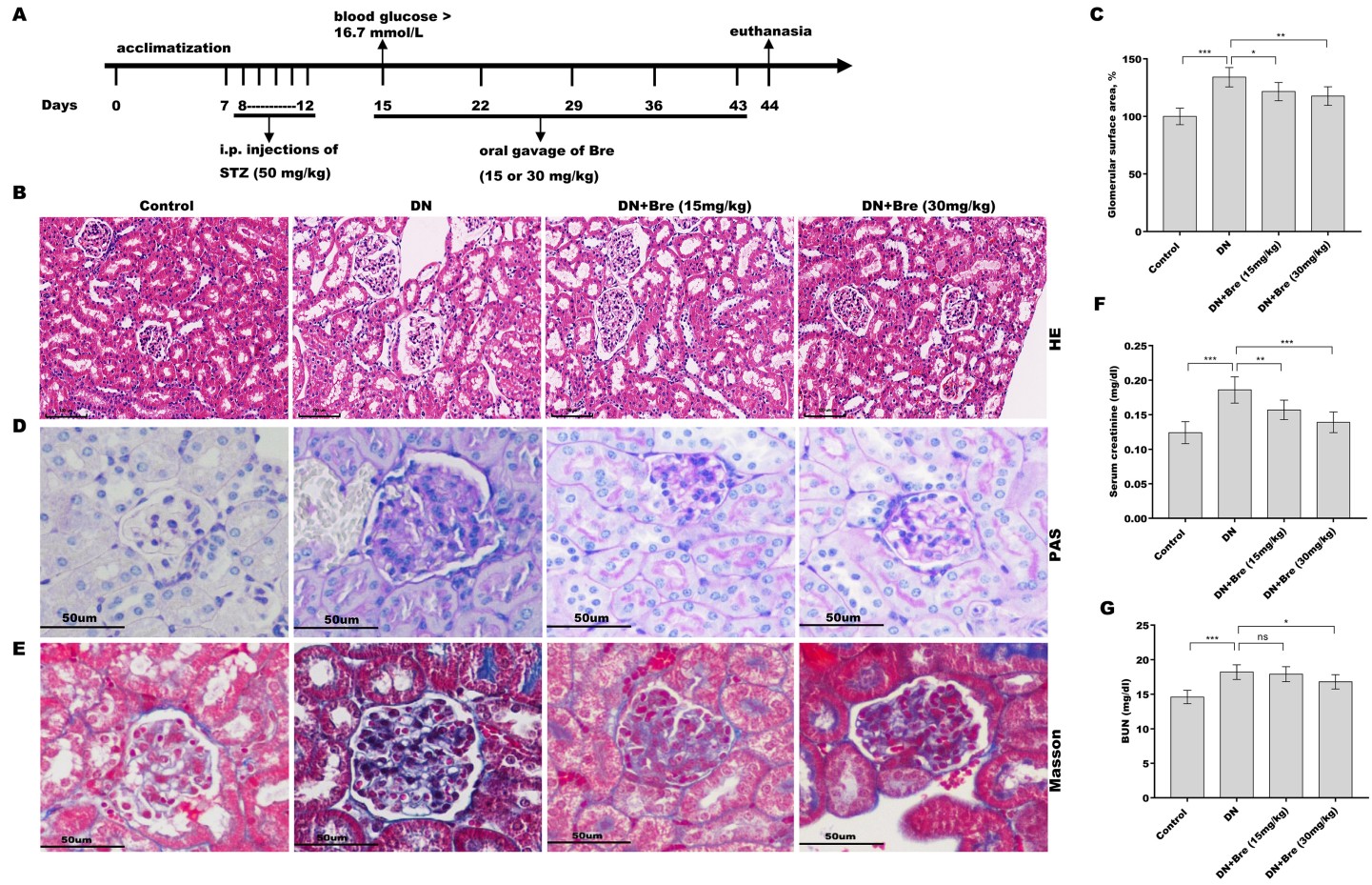

**Figure 1 Bre decreased glomerular injury in DN mice.** (A) The establishment of diabetic mice model and treatment procedure. After being treated with Bre (15 or 30 mg/kg) for 4 weeks, the mice were euthanized, and renal tissues were collected for HE (B, scale bar = 100 μm), PAS (D, scale bar = 50 μm), and Masson (E, scale bar = 50 μm) staining. (C) The glomerular surface area was measured in accordance with the results of HE staining. After being treated with Bre (15 or 30 mg/kg) for 4 weeks, the mice were euthanized, and blood samples were collected to assess serum creatinine (F) and BUN (G) levels. n = 7. $^{*}p < 0.05$, $^{**}p < 0.01$, $^{***}p < 0.001$.

## CCK-8 assay

MPC5 cells were cultured in 96-well plates (3,000 cells per well) and treated with HG and other reagents (50 or 100 μM Bre, 10 μM MCC950, or 40 μM Ac-YVAD-cmk). Forty-eight hours later, the cells were treated with CCK-8 (10 μL, Abcam, Burlingame, CA, USA) for 1 h, and then the absorbance was measured at 450 nm with a microplate reader (Bio-Rad, Hercules, CA, USA).
## Western blot analysis

Total protein was isolated from MPC5 cells using RIPA buffer (Thermo Fisher Scientific, Waltham, MA, USA) and quantitated using a BCA protein quantification kit (Abcam, Cambridge, UK), and the proteins were separated *via* 10% SDS–PAGE. Then, the proteins were transferred to PVDF membranes (Roche, Basel, Switzerland). After being sealed with 5% nonfat milk and washed three times with TBST, the membranes were incubated with primary antibodies against Bax (1:1,500, ab182733; Abcam, Cambridge, UK), Bcl-2 (1:2,000, ab182858; Abcam, Cambridge, UK), α-SMA (1:400, MA5-14084; Thermo Fisher Scientific, Waltham, MA, USA), podocin (1:800, PA5-79757; Thermo Fisher Scientific, Waltham, MA, USA), synaptopodin (1:1,000, ab259976; Abcam, Cambridge, UK), p-p65 (1:800, MA5-15160), NLRP3 (1:800, ab263899), cleaved caspase-1 (1:1,000, PA5-99390; Thermo Fisher Scientific, Waltham, MA, USA), GSDMD (1:900, ab209845; Abcam, Cambridge, UK), and β-actin (1:8,000, MA1-140; Thermo Fisher Scientific, Waltham, MA, USA) for 1 h at room temperature. After being washed three times with TBST, the membranes were incubated with HRP-conjugated secondary antibodies (1:7,000) for 1 h, and then the immunoblots were visualized using an ECL kit (Roche, Basel, Switzerland) and quantitated using ImageJ software (Bethesda, Rockville, MD, USA).

## IF analysis

MPC5 cells were fixed with 4% PFA for 15 min, permeabilized with 0.5% Triton X-100 for 25 min, sealed with normal donkey serum for 60 min, and then incubated with primary antibodies against α-SMA (1:100, MA5-14084; Thermo Fisher Scientific, Waltham, MA, USA), synaptopodin (1:100, ab259976; Abcam, Cambridge, UK), and p65 (1:300, ab32536; Abcam, Cambridge, UK) overnight at 4 °C. After being washed three times with PBST, the slides were incubated with goat anti-rabbit antibodies (1:1,500, A27039; Invitrogen, Waltham, MA, USA) for 2 h in the dark. Nuclei were visualized by staining with DAPI (Solarbio, Shanghai, China). Images were obtained using a DMI4000B fluorescence microscope (Leica, Wetzlar, Germany).

## TUNEL assay

MPC5 cells were treated with HG and other reagents (50 or 100 μM Bre, 10 μM MCC950, or 40 μM Ac-YVAD-cmk) for 48 h and then subjected to TUNEL staining (Solarbio, Beijing, China) to assess apoptosis. TUNEL-positive cells were observed with a DMI4000B fluorescence microscope.

## Serum creatinine and blood urea nitrogen (BUN) levels

Serum samples were collected, and serum creatinine and BUN levels were measured with an automated biochemical analyser (PUZS-600A/B; PERLONG, Beijing, China).

## Histological analysis

Renal tissues were collected, fixed with 4% PFA, embedded in paraffin, and cut into 4 μm-thick sections. The sections were stained with haematoxylin and eosin (H&E), periodic-acid-Schiff (PAS), and Masson's trichrome by staining kits in accordance with the manufacturer's instructions (Solarbio, Beijing, China).

## LDH assay

LDH was measured using an LDH assay kit (ab65393; Abcam, Cambridge, UK). The absorbance was measured at 450 nm with a microplate reader (Bio-Tek, Winooski, VT, USA).

## Statistical analysis

The data are displayed as the mean ± SD from at least three independent replicates and analysed using SPSS 20 software (IBM, Armonk, NY, USA). Student's $t$ test was used for intergroup comparisons and one-way ANOVA (Scheffé test) was used for multi-group comparison. Data were considered statistically significant when the $P$ value was less than 0.05.

# RESULTS

## Bre decreased glomerular injury in DN mice

To investigate the effect of Bre on the regulation of renal function in DN mice, the mice were i.p. injected with STZ and then treated with Bre by intragastric administration. As shown in Figs. 1B and 1C, the glomerular surface area was prominently increased in DN mice, whereas Bre treatment effectively inhibited this increase. PAS and Masson staining revealed the proliferation of the mesangial matrix and glomerular fibrosis in DN mice, and these effects were reversed by Bre (Figs. 1D and 1E). Serum creatinine and BUN levels, which are known biomarkers of renal function, were further assessed after treatment with Bre. Figures 1F and 1G show that serum creatinine and BUN levels were significantly elevated in DN mice, and these effects were blocked by Bre treatment. These results demonstrate that Bre alleviates glomerular injury in DN mice.

## Bre decreased HG-induced podocyte apoptosis *in vitro*

Then, the effects of Bre on the regulation of podocyte viability and apoptosis were assessed *in vitro*. Figure 2A shows that podocyte viability was significantly decreased after HG treatment, and Bre effectively restored cell viability in a dose-dependent manner. The results of TUNEL staining showed that HG induced marked podocyte apoptosis, and these effects were reversed by Bre (Figs. 2B and 2C). Furthermore, there was a significant increase in Bax protein expression and a reduction in Bcl-2 protein expression in podocytes under HG conditions compared with the low glucose control, and these effects were blocked by Bre (Fig. 2D). Bre treatment also inhibited HG-induced increase in reactive oxygen species (ROS) levels in podocytes (Fig. 2E).

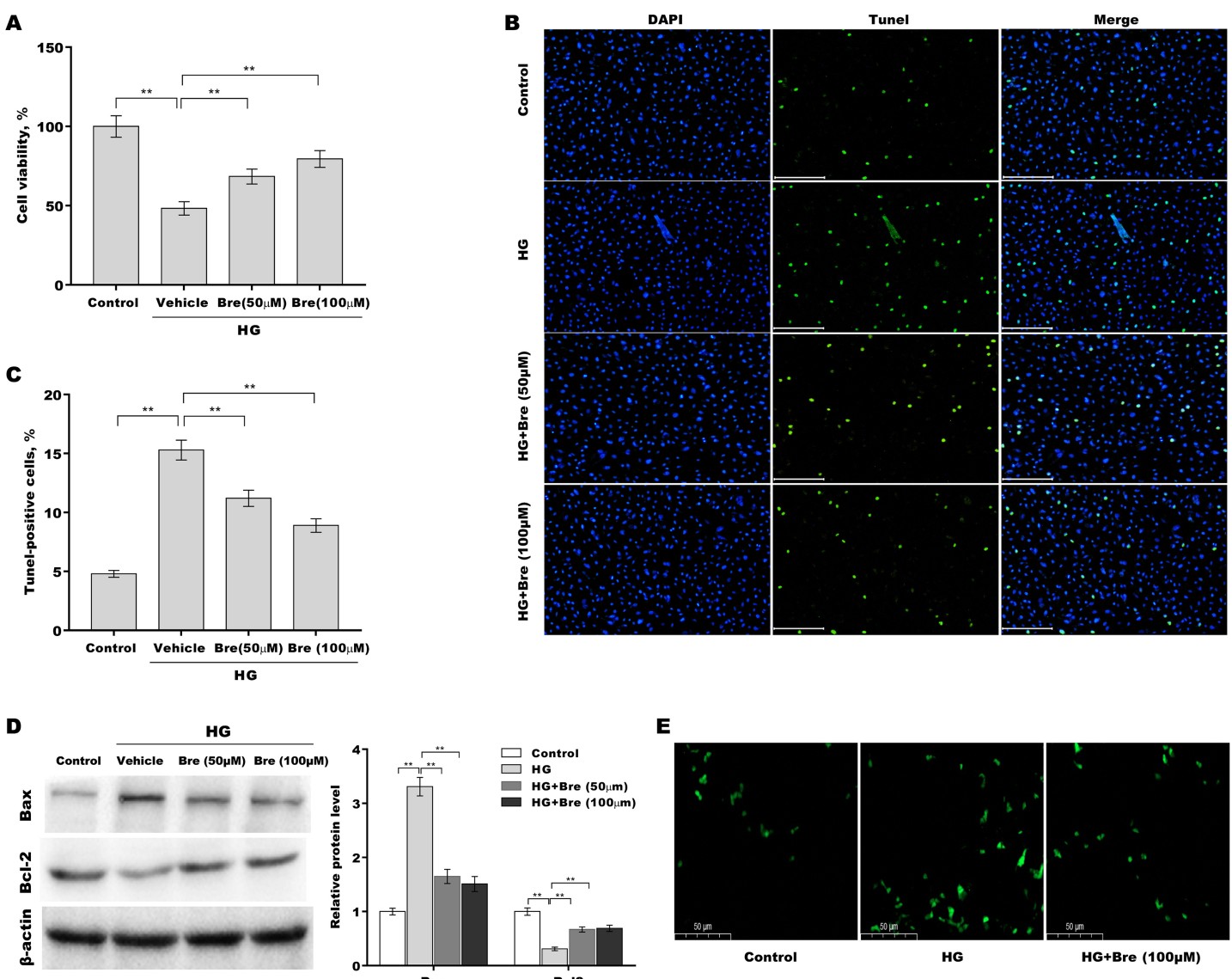

**Figure 2 Bre decreased HG-induced podocyte apoptosis *in vitro*.** After being treated with HG and Bre (50 or 100 µM) for 48 h, MPC5 cell viability (A) and apoptosis (B and C) were assessed using CCK-8 assays and TUNEL staining, respectively. (D) After being treated with HG and Bre (50 or 100 µM) for 48 h, Bax and Bcl-2 protein levels in MPC5 cells were assessed by western blot analysis. (E) Intracellular ROS levels were assessed in MPC5 cells after treatment with HG and Bre (100 µM). **$p < 0.01$.

## Bre decreased HG-induced podocyte injury *in vitro*

The biological roles of Bre in regulating HG-induced podocyte injury were next evaluated by assessing α-SMA, podocin, and synaptopodin expression in podocytes. As shown in Fig. 3A, α-SMA mRNA levels were markedly elevated, and podocin and synaptopodin mRNA levels were downregulated under HG conditions compared with the low glucose control; these changes were significantly blocked by Bre treatment. Consistent with the qRT‑PCR results, western blot analysis revealed similar effects of Bre on the regulation of α-SMA, podocin, and synaptopodin protein expression in HG-treated podocytes (Figs. 3B and 3C). IF staining revealed that HG treatment increased α-SMA levels and decreased

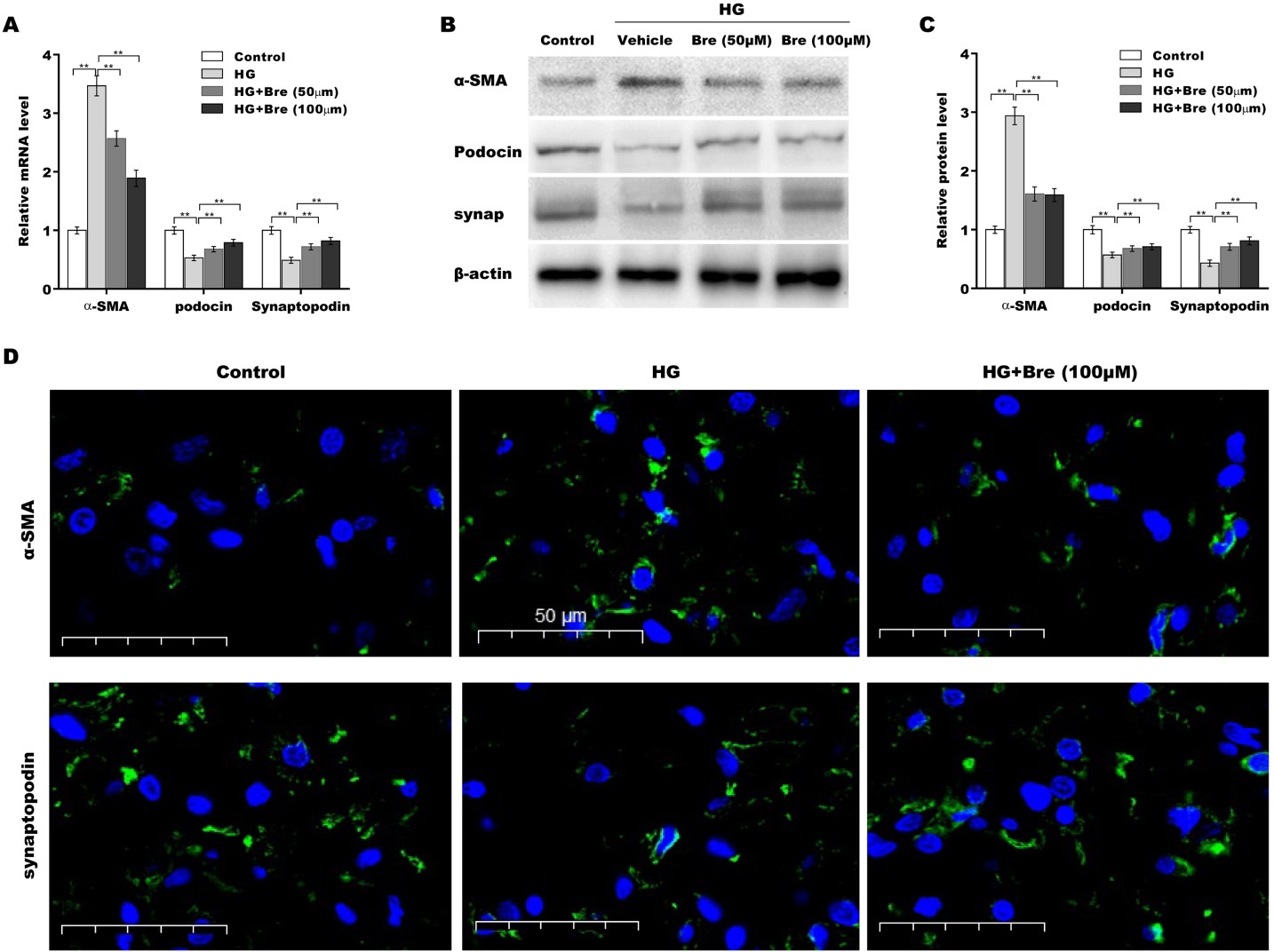

**Figure 3** **Bre decreased HG-induced podocyte injury *in vitro*.** After being treated with HG and Bre (50 or 100 μM) for 48 h, the mRNA (A) and protein levels (B and C) of α-SMA, podocin, and synaptopodin were assessed in MPC5 cells using qRT–PCR and western blot analysis, respectively. (D) IF analysis of α-SMA and synaptopodin in MPC5 cells after treatment with HG and Bre (50 or 100 μM) for 48 h. **$p < 0.01$.

synaptopodin levels in podocytes, while Bre effectively inhibited these changes (Fig. 3D). These data demonstrate that Bre alleviates HG-induced podocyte injury *in vitro*.

## Bre decreased NF-κB/NLRP3-mediated pyroptosis in HG-treated podocytes *in vitro*

Given the crucial role of NF-κB signalling and NF-κB-mediated NLRP3 inflammasome activation in podocyte injury (*Ke et al., 2021*; *Lv et al., 2022*; *Xu et al., 2021*), the effects of Bre on the regulation of NF-κB/NLRP3 signalling activation in podocytes were next investigated. Podocytes were treated with HG and Bre, and then phosphorylated p65 (p-p65) levels were assessed by western blot analysis. Figures 4A and 4B show that p-p65 levels were markedly increased in podocytes under HG stress, and these effects were

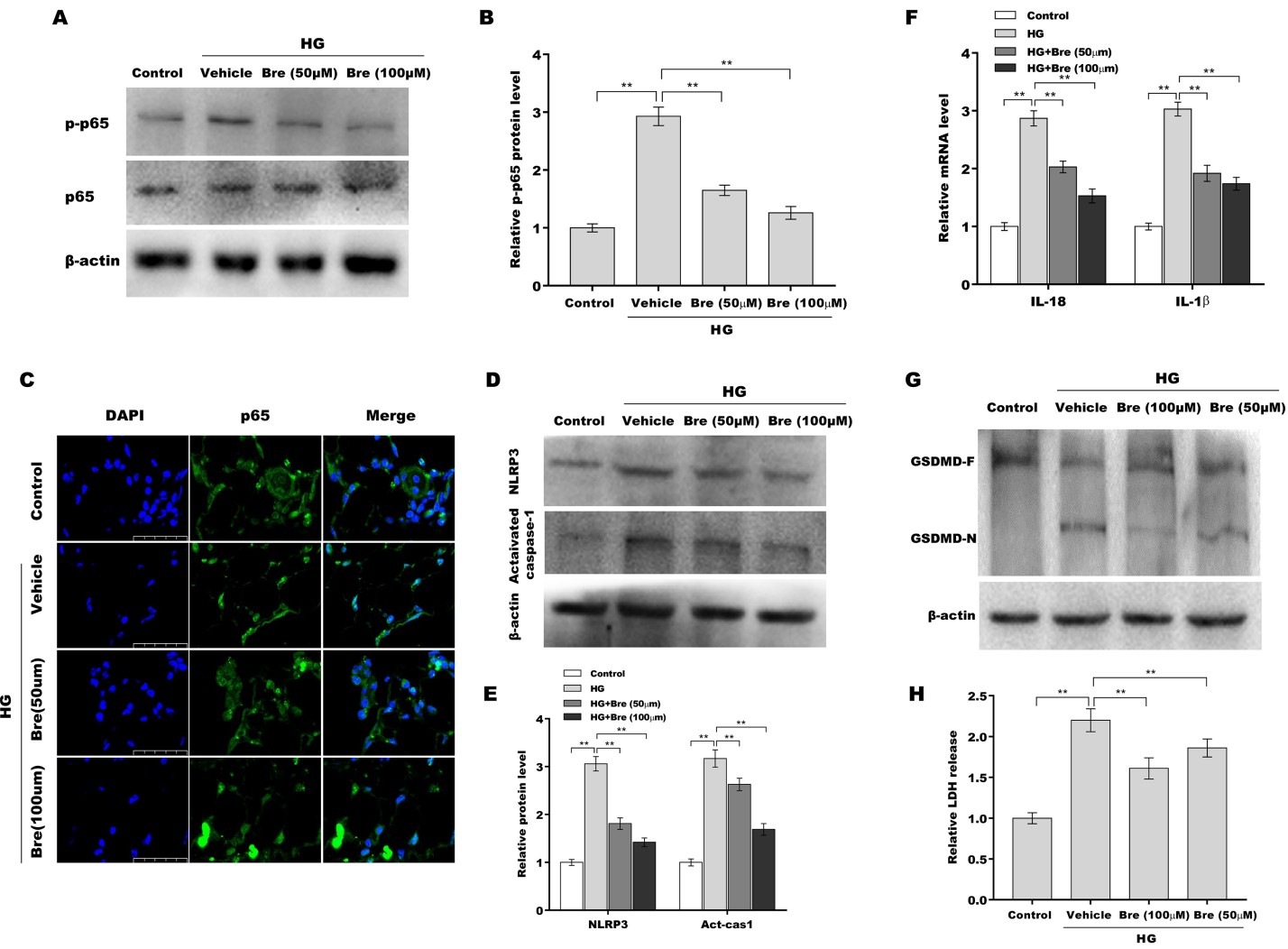

**Figure 4  Bre decreased NF-κB/NLRP3-mediated pyroptosis in HG-treated podocytes *in vitro*.** (A and B) After being treated with HG and Bre (50 or 100 μM) for 48 h, the protein levels of p-p65 were assessed in MPC5 cells by western blot analysis. (C) IF analysis was carried out to assess the nuclear translocation of p65 in MPC5 cells after being treated with HG and Bre (50 or 100 μM) for 48 h. (D and E) After being treated with HG and Bre (50 or 100 μM) for 48 h, the protein levels of NLRP3 and activated caspase 1 were assessed in MPC5 cells by western blot analysis. (F) qRT–PCR analysis of IL-18 and IL-1β mRNA levels in MPC5 cells. (G) Western blot analysis of GSDMD protein levels in MPC5 cells. (H) ELISA analysis of LDH release in MPC5 cells. **$p < 0.01$.                                               

significantly reversed by Bre, indicating that Bre could inhibit HG-induced NF-κB activation. To further verify this hypothesis, nuclear translocation of p65 was assayed in podocytes treated with HG and Bre. Figure 4C shows that Bre suppressed HG-induced p65 nuclear translocation in podocytes. Furthermore, NLRP3 and activated caspase-1 protein levels were prominently increased in podocytes under HG conditions compared with the control, and Bre alleviated HG-induced NLRP3 inflammasome activation (Figs. 4D and 4E). Thus, Bre further decreased the HG-induced release of IL-18 and IL-1β (Fig. 4F). Bre also reduced HG-induced IL-1β production in cell supernatant after treatment with Bre (Fig. S1). Subsequently, NLRP3-mediated pyroptosis in HG-treated podocytes was investigated by assessing GSDMD cleavage and LDH release. As expected, HG treatment

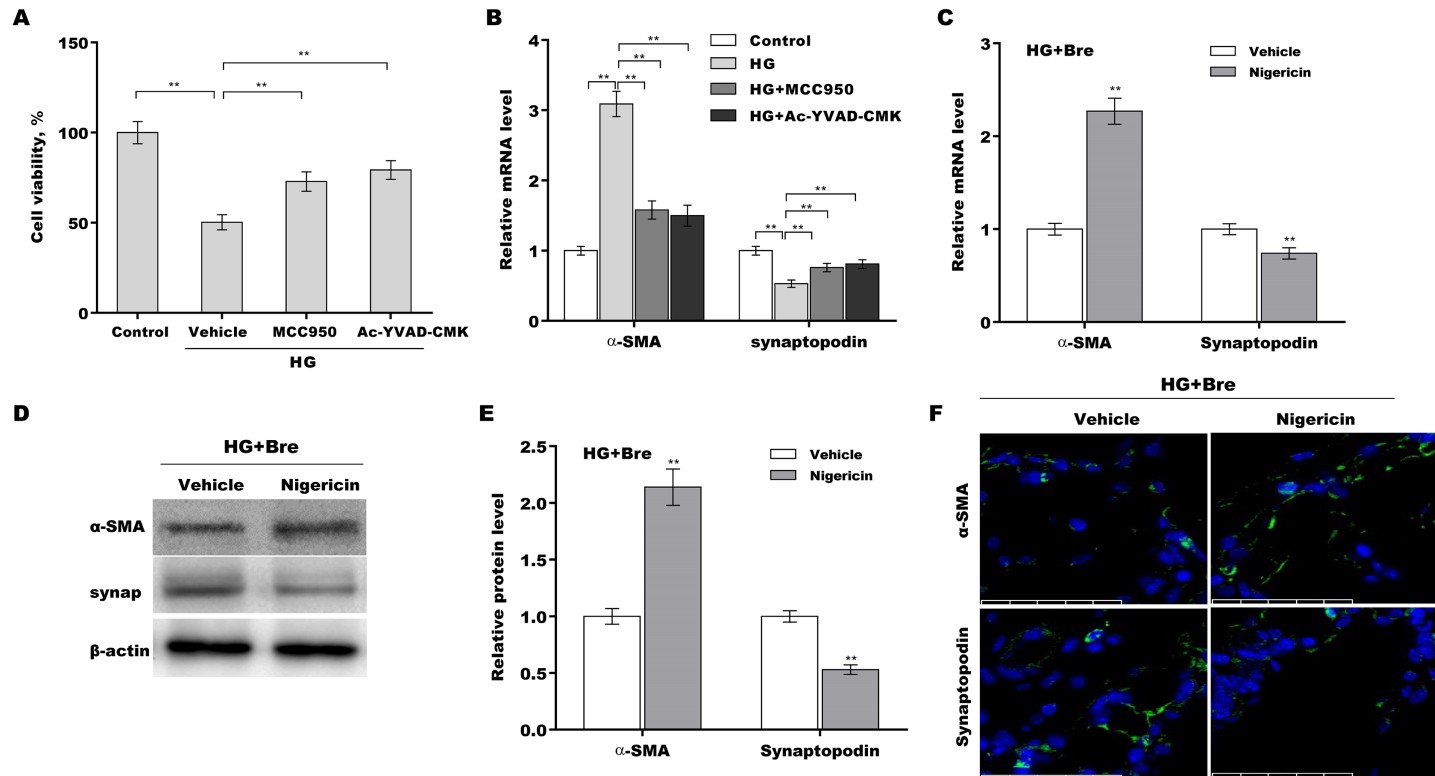

**Figure 5 Bre decreased HG-induced podocyte injury by inhibiting NLRP3-mediated pyroptosis *in vitro*.** (A) After being treated with HG, MCC950 (10 μM), and Ac-YVAD-cmk (40 μM) for 48 h, MPC5 cell viability was assessed using CCK-8 assays. (B) After being treated with HG, MCC950 (10 μM), and Ac-YVAD-cmk (40 μM) for 48 h, the mRNA levels of α-SMA and synaptopodin in MPC5 cells were assessed using qRT-PCR. qRT-PCR (C), western blotting (D and E), and IF (F) analysis were carried out to assess α-SMA and synaptopodin levels in MPC5 cells after treatment with nigericin (10 μM) for 48 h in the presence of HG and Bre. $^{**}p < 0.01$.

resulted in obvious cleavage of GSDMD and elevated levels of the GSDMD-N fragment in podocytes, and these changes were blocked by Bre (Fig. 4G). Bre also decreased the HG-induced release of LDH (Fig. 4H). These data suggest that Bre alleviates NF-κB/NLRP3-mediated pyroptosis in HG-treated podocytes.

## Bre decreased podocyte injury by inhibiting NLRP3-mediated pyroptosis *in vitro*

The role of NLRP3-mediated pyroptosis in podocyte injury was next investigated using MCC950 (NLRP3 inhibitor) (*Tang et al., 2020*), Ac-YVAD-CMK (caspase-1 inhibitor) (*Wang et al., 2022b*), and nigericin (NLRP3 activator) (*Jian et al., 2020*; *Mo et al., 2021*). Podocytes were treated with HG in the presence or absence of MCC950, Ac-YVAD-CMK, or nigericin, and then podocyte injury was evaluated. HG treatment caused a prominent decrease in podocyte viability, and these effects were reversed by MCC950 or Ac-YVAD-CMK (Fig. 5A). MCC950 or Ac-YVAD-CMK decreased α-SMA levels and increased synaptopodin levels in HG-treated podocytes (Fig. 5B), indicating that NLRP3-mediated pyroptosis exerted crucial effects on HG-induced podocyte injury. To further examine whether Bre alleviated podocyte injury by inhibiting NLRP3 inflammasome activation, podocytes were treated with HG plus Bre in the presence or absence of nigericin, and podocyte injury was evaluated. Figures 5C–5E

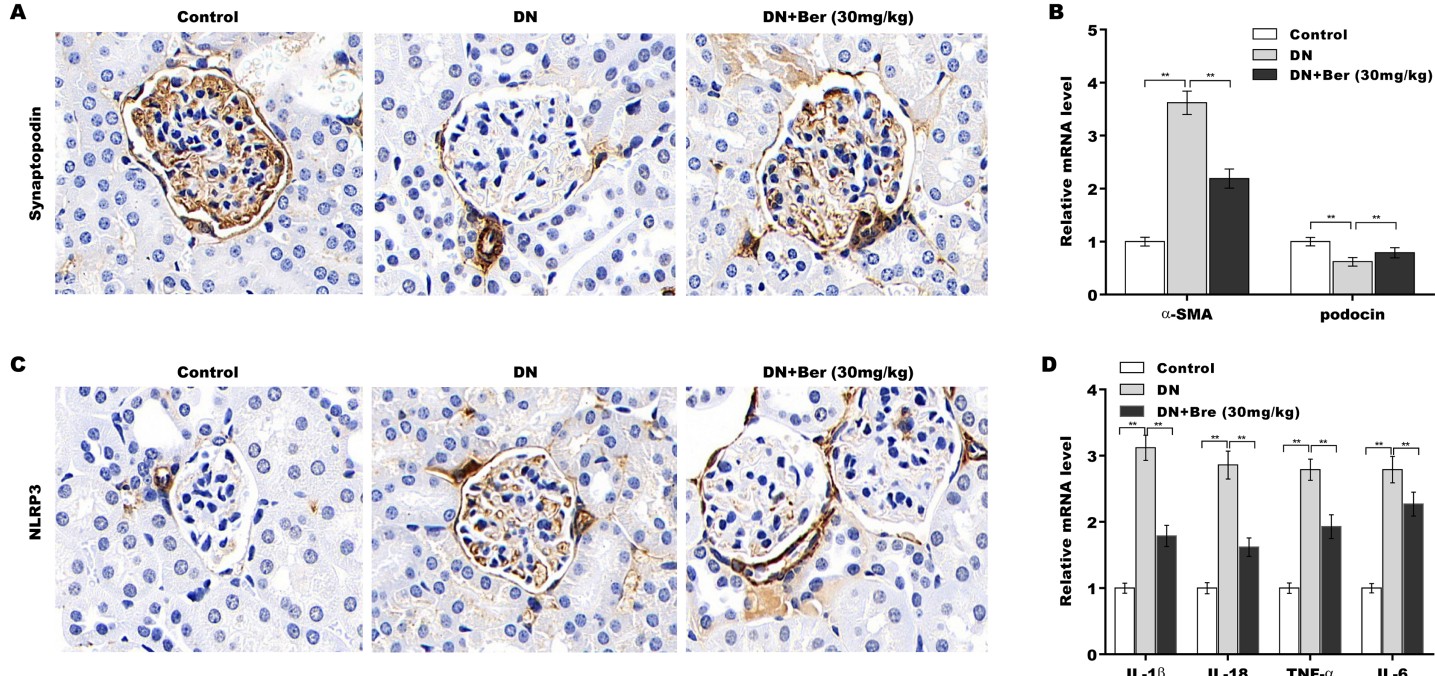

**Figure 6 Bre inhibited NLRP3 activation and decreased podocyte injury in the renal tissues of DN mice.** (A) After being treated with Bre (30 mg/kg) for 4 weeks, the mice were euthanized, and renal tissues were collected to assess synaptopocin expression by IHC analysis. (B) qRT-PCR analysis of α-SMA and podocin mRNA levels in the renal tissues of control mice, DN mice, and Bre-treated mice. (C) After being treated with Bre (30 mg/kg) for 4 weeks, the mice were euthanized, and renal tissues were collected to assess NLRP3 expression by IHC analysis. (D) qRT-PCR analysis of IL-1β, IL-18, TNF-α, and IL-6 mRNA levels in the renal tissues of control mice, DN mice, and Bre-treated mice. $^{**}p < 0.01$.

shows that compared with the vehicle, nigericin increased α-SMA expression and decreased synaptopodin expression at the mRNA and protein levels in the presence of HG plus Bre. IF staining revealed similar results (Fig. 5F).

## Bre inhibited NLRP3 activation and decreased podocyte injury in the renal tissues of DN mice

Finally, the effects of Bre on inhibiting NLRP3 activation and decreasing podocyte injury were assessed in the renal tissues of DN mice. The IHC results revealed that synaptopodin expression was decreased in the renal tissues of DN mice, and this effect was reversed by Bre (Fig. 6A). Moreover, α-SMA levels were increased and podocin levels were decreased in DN mice, and these alterations were blocked by Bre (Fig. 6B). The IHC results further revealed that NLRP3 expression was elevated in the renal tissues of DN mice, and Bre effectively inhibited this increase (Fig. 6C). As a result, Bre attenuated proinflammatory cytokine production in the renal tissues of DN mice (Fig. 6D).

## DISCUSSION

The pathogenesis of DN is extremely complicated and poorly understood.
The conventional treatment for early DN generally involves lowering blood sugar and blood pressure and blocking the RAAS (*Chang et al., 2020*; *Choudhury, Tuncel & Levi, 2010*). Unfortunately, DN frequently progresses into glomerulosclerosis, kidney fibrosis,

and ESRD (*Dong et al., 2022*). Given the renoprotective roles of Bre in diabetic mice, we investigated whether Bre exerts its renoprotective effects by alleviating HG-induced podocyte injury. We demonstrated that (i) Bre alleviated glomerular injury in DN mice, (ii) Bre alleviated HG-induced podocyte apoptosis and podocyte injury, (iii) Bre alleviated NF-κB/NLRP3-mediated pyroptosis in HG-treated podocytes, and (iv) Bre alleviated HG-induced podocyte injury by inhibiting NLRP3-mediated pyroptosis.

Recent studies have revealed the protective effects of Bre against renal injury (*Liu et al., 2016b*), cerebral ischaemia–reperfusion injury (*Chen et al., 2022*), liver injury (*Lan et al., 2022*; *Liu et al., 2018*), and myocardial damage (*Wang et al., 2009*) through various mechanisms, such as inflammation, ROS production, and endoplasmic reticulum stress (*Schena & Gesualdo, 2005*; *Wolf, 2004*). Although inflammation is a protective response to infection and tissue injury, an excessive inflammatory response results in long-lasting tissue injury (*Liu et al., 2017*). NF-κB plays a critical role in orchestrating multiple aspects of the inflammatory response. Furthermore, NF-κB signalling is an inevitable prerequisite for the activation of the NLRP3 inflammasome (*Boaru et al., 2015*), which has vital effects on the production of proinflammatory cytokines and the initiation of pyroptosis (*Yu et al., 2022*). Emerging evidence has demonstrated that various active components derived from traditional Chinese herbal medicines exhibit renoprotective effects by inhibiting the NLRP3 inflammasome. Fucoidan, an active component in *Laminaria japonica*, attenuates renal fibrosis by inhibiting NLRP3-dependent pyroptosis in podocytes (*Wang et al., 2022a*). As a natural dietary flavonoid in various fruits, fisetin alleviates podocyte injury in DN mice by inhibiting NLRP3 inflammasome activation (*Dong et al., 2022*). Bre is a flavonoid extract from *Erigeron breviscapus* and exerts renoprotective effects on diabetic rats (*Qi et al., 2006*; *Xu et al., 2013*). Here, the role of Bre in regulating podocyte injury in DN mice was further investigated. We demonstrated that Bre improved renal function in diabetic mice. Furthermore, Bre increased podocyte viability and decreased HG-induced podocyte apoptosis and injury. Mechanistically, Bre attenuated HG-induced NF-κB/ NLRP3 activation, resulting in decreased podocyte pyroptosis.

Although the role of Bre in inhibiting NLRP3 inflammasome activation has been revealed, additional mechanistic studies are necessary to provide further insights into the underlying mechanisms by which Bre inhibits NLRP3 activation. Bre possesses antioxidant activities (*Zhang et al., 2022b*), and ROS are crucial for triggering NLRP3 inflammasome formation and activation (*Tschopp & Schroder, 2010*). Therefore, it would be worthwhile to examine whether Bre represses NLRP3 activation in a ROS-dependent manner, as well as NF-κB signalling. In addition, podocyte injury and excessive mesangial cell proliferation are the two major characteristics of DN (*Nagata, 2016*; *Rousseau et al., 2022*; *Thomas & Ford Versypt, 2022*). It is important to investigate whether Bre regulates NLRP3 activation in mesangial cells in addition to podocytes.

## CONCLUSION

Bre alleviates HG-induced podocyte injury by inhibiting NF-κB/NLRP3-mediated pyroptosis in DN mice.

### Funding

This study was funded by the Natural Science Foundation of Shanghai (grant number 20ZR1451600) and the Shanghai Municipal Health Bureau Project (grant number 201940439). The funders had no role in study design, data collection and analysis, decision to publish, or preparation of the manuscript.

### Grant Disclosures

The following grant information was disclosed by the authors:
Natural Science Foundation of Shanghai: 20ZR1451600.
Shanghai Municipal Health Bureau Project: 201940439.

### Competing Interests

All authors declare that they have no competing interests.

### Author Contributions

- Linlin Sun conceived and designed the experiments, performed the experiments, analyzed the data, authored or reviewed drafts of the article, and approved the final draft.
- Miao Ding performed the experiments, analyzed the data, prepared figures and/or tables, authored or reviewed drafts of the article, and approved the final draft.
- Fuhua Chen performed the experiments, prepared figures and/or tables, authored or reviewed drafts of the article, and approved the final draft.
- Dingyu Zhu performed the experiments, prepared figures and/or tables, authored or reviewed drafts of the article, and approved the final draft.
- Xinmiao Xie performed the experiments, analyzed the data, prepared figures and/or tables, authored or reviewed drafts of the article, and approved the final draft.

### Data Availability

The raw measurements are available in the Supplemental Files.

### Supplemental Information

Supplemental information for this article can be found online at http://dx.doi.org/10.7717/peerj.14826#supplemental-information.

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
