# Peer review of "Breviscapine alleviates podocyte injury by inhibiting NF-κB/NLRP3-mediated pyroptosis in diabetic nephropathy"

_PeerJ, doi:10.7717/peerj.14826_

## Round 0.1 · original submission · Major Revisions

The Reviewers have highlighted some important points that require attention in the manuscript. Please address all the concerns addressed by the Reviewers. Attention is required for in vitro experiments demonstrating NLRP3 inflammasome activation in vitro- Data for cleaved IL-1b or IL-18 in cell culture supernatant needs to be analyzed by ELISA and Western blot. Changes in GasD is one of the important parameters and may not necessarily demonstrate the full effect of NLRP3 inflammasome activation. Please analyze levels of IL-1b, IL-18, Casp-1, Gas D etc. in renal tissues in addition to NLRP3 in Figure 6. Additionally, degradation/reduction in levels of NLRP3 protein was observed in vitro and in vivo. Is it normal during inflammasome activation? Please explain.

Reviewer 1 ·

Basic reporting

There are lots of grammatical errors throughout this manuscript. Please re-check and rewrite this manuscript.

Experimental design

In this study authors have investigated how breviscapine (Bre) acts as a reno protective agent in diabetic mice. In this study they have shown that Bre prevents HG-induced podocyte injury and recovers renal function via NF-kB/NLRP3-mediated signaling pathways in vitro and in STZ induced diabetic mice. This is an interesting study and before considering for publication the authors should go through the following comments.
1. Authors have shown that HG induces podocytes apoptotic, NF-kB and NLRP3 inflammasome pathway. Reactive oxygen species (ROS) has a major factor in inducing those pathways. But authors don’t show any data about ROS. They should measure HG induced ROS activation in podocytes, at least in cultured cells by using immunofluorescence (IF) or flow cytometry.
2. Authors have shown Bcl2 level (Fig. 2, D) via western blotting. But I don’t see the differences between Bcl-2 levels in HG induced and Bre treated cell. Authors should repeat this experiment.
3. Authors have shown IL-1b mRNA expression in HG-induced and Bre-treated mouse podocyte, MPC5 cells. They should check released cleaved IL-1b in media by ELISA.

Validity of the findings

NA

Additional comments

There are lots of grammatical errors throughout this manuscript. Please re-check and rewrite this manuscript.

Annotated reviews are not available for download in order to protect the identity of reviewers who chose to remain anonymous.

·

Basic reporting

The author meticulously provides a literature review and background information. However, the author didn't discuss the available treatments as well as their limitations. Manuscript professionally written.

Author provides raw data and figures. However method used for the microscopic image quantification should be explain in the methods sections.

Experimental design

The Author well explained the research question and experimental design. The Author established the knowledge gap and did a rigorous investigation to answer all the questions. However, the author couldn't explain the connection between NLRP3 and Diabetic Nephropathy. Are there any studies that established the role of NLRP3 in Diabetic nephropathy that need further explanation with evidence? Your introduction needs more detail. I suggest you improve the description in lines 55-56 to provide more justification for your study (specifically, NLRP3 role in diabetic nephropathy).
The author should also provide either the reference or EC50 data of Breviscapine. The author provides methods with sufficient details and information. However, the statistics and multiple comparisons used in the study need further clarification, such as in Fig 2E, Fig 3A, Fig 3C, Fig 4E, Fig 5B, 5C, 5E, 6B, and 6D. Spacing and formatting need to be carefully checked, such as line 238.

Validity of the findings

The author provides an ample amount of data that suggests the effect of Breviscapine on diabetic nephropathy via NF-kB signaling however, the author couldn't able to provide enough evidence that provides the primary molecular target of Breviscapine in Nf-kB signaling. As Jiang et al., 2016 explain the role of Bre in AKT and MAPK signaling that resulted in NF-kB signaling, further molecular mechanisms need to be established to prove the author's claims.

·

Basic reporting

The study by Sun et al., investigate Breviscapine, a flavonoid extract amelioration of HG-induced podocyte injury that improves renal function in diabetic mice by suppressing NF-κB/NLRP3-mediated pyroptosis. In this study the authors showed that Breviscapine effectively increased inhibited HG-induced NF-κB signaling activation and subsequently decreased NLRP3 inflammasome activation, resulting in a decreased pyroptosis. This study is appropriately conducted but there is much room for improvement. The parameters in the study are properly evaluated and the manuscript is well written. Methods and results are discussed appropriately in the manuscript. While this is the addition of more information to similar types of studies which somewhat already have been conducted, this information paves a closer way to understanding NF-κB/NLRP3-mediated pyroptosis and its possible therapeutic interventions. The topic is clinically important, but there are critical problems that need to be addressed:
• Study introduction is poorly written, in general, it’s nice to end an introduction section with a few comments about the aim and hypothesis behind the study instead of referring to other studies, to help readers understand what’s coming; please consider adding some relevant remarks here.

Experimental design

• The experimental strategies design and data reported are very poor and not very convincing. It is difficult to understand exactly what type of diabetic mouse model they used in this study. Here they have mentioned diabetes mellitus but whether it is type 2 diabetes or type 1 diabetes is unclear. With the type of doses, it looks like a low-dose STZ-induced T1D mouse model should provide an appropriate reference for this. The authors should provide an explanation of why they have chosen a low-dose STZ diabetic mouse model.
• It is very unclear the doses Treatment: What are the parameters for the treatment doses? Why 30mg/kg? Did the author test dose of treatment earlier? The reference provided by the authors has not used the doses of 10 mg/kg/d for 4 weeks (Jiang et al. 2016; Lan et al. 2022), Please discuss or provide the correct reference (Page no 5, lines 86-88) that could help readers.
• Authors should explain the composition of Breviscapine extract, how it has been extracted, and the preparation of Breviscapine extract should be part of the methodology.
• Mouse model of diabetes. Why did the authors prefer a low-dose STZ-induced mouse model of diabetes rather than other available models of diabetes? Doe this model perfectly represent stable glucose levels and does not revert disease conditions upon normal diet feeding? The authors should justify their choice of model. It can be a part of the discussion.
• Entire methodology section needs to be rewritten in detailed procedure with appropriate references.
• How many repeat experiments have been performed (in vivo and in vitro assays)? At least one repeat experiment is required and has to be shown.
• It would be of considerable interest to know how the disease progresses after the curtailment of treatment. Does blood glucose stay down, improve insulin sensitivity, or does the disease begin to worsen? in other words longevity for the effect. Such data would give an indication of what form treatment might take if these extracts were to be developed as therapeutic agents.
• It is advisable to describe the treatment procedures schematically, it will be easy to grasp for the readers.

Validity of the findings

• It would be of considerable interest to know how the disease progresses after the curtailment of treatment. Does blood glucose stay down, improve insulin sensitivity, or does the disease begin to worsen? in other words longevity for the effect. Such data would give an indication of what form treatment might take if these extracts were to be developed as therapeutic agents.
• What would be the thinking for the route of administration (i.p), however, it wouldn't be convenient in an actual clinical setting.

Additional comments

• The discussion needs minor revision. How this strategy is better or comparable to the success achieved in other studies is lacking?
• It is advised that the authors should define abbreviations at least in the first instance.
• There are a lot of definite and indefinite articles missing along with some grammatical errors, also please correct the manuscript for undefined spaces, an editor is required to correct these.
• Number of spelling mistakes/typo errors.
• Citations and references must be in the correct and uniform format, Italics missing in some places.

---

## Round 0.2 · accepted · Accept

Dear Dr. Sun,

Thank you for submitting the revised version of the manuscript titled, "Breviscapine alleviates podocyte injury by inhibiting NF-κB/NLRP3-mediated pyroptosis in diabetic nephropathy". After reviewing the revised version of the manuscript, the reviewers have recommended the manuscript to be considered for publication. Congratulations!

I have further reviewed the revised version and recommend the manuscript for publication.

Best wishes,
Himangshu Sonowal

Reviewer 1 ·

Basic reporting

no comment

Experimental design

no comment

Validity of the findings

no comment

Additional comments

no comment

·

Basic reporting

no comment

Experimental design

no comment

Validity of the findings

no comment

Additional comments

no comment

·

Basic reporting

The revised version of the manuscript is very much improved. I am very much convinced with the authors' clarifications raised by me and other reviewers. It is a very informative manuscript in terms of knowledge in the field.

Experimental design

The schematic for the experimental treatment is very informative and easy to grasp for the readers.

Validity of the findings

No comments

Additional comments

No comments